Prediction of perinatal depression among women in Pakistan using Hybrid RNN-LSTM model

http://orcid.org/0000-0002-7270-7238 Zafar Amna 1
Wasim Muhammad 2
http://orcid.org/0000-0002-5348-9132 Akram Beenish Ayesha 3
Riaz Maham 1
http://orcid.org/0000-0002-3394-6762 Pires Ivan Miguel 4
http://orcid.org/0000-0002-4383-0472 Coelho Paulo Jorge 5 6 paulo.coelho@ipleiria.pt
1 Department of Computer Science, University of Engineering and Technology Lahore , Lahore , Pakistan
2 Department of Computer Science, University of Management & Technology, Lahore , Lahore (Sialkot Campus) , Pakistan
3 Department of Computer Engineering, University of Engineering and Technology Lahore , Lahore , Pakistan
4 Instituto de Telecomunicações, Escola Superior de Tecnologia e Gestão de Águeda, Universidade de Aveiro , Águeda , Portugal
5 Institute for Systems Engineering and Computers at Coimbra, University of Coimbra , Coimbra , Portugal
6 School of Technology and Management, Polytechnic University of Leiria , Leiria , Portugal
Cirillo Stefano
Electronic publication date: 2025 Feb 26
Publication date: 2025
Volume: 11
Electronic Location ID: e2673
Received 2024 May 24; Accepted 2025 Jan 7
Copyright: © 2025 Zafar et al.
Copyright year: 2025
Copyright holder: Zafar et al.
License: This is an open access article distributed under the terms of the Creative Commons Attribution License, which permits unrestricted use, distribution, reproduction and adaptation in any medium and for any purpose provided that it is properly attributed. For attribution, the original author(s), title, publication source (PeerJ Computer Science) and either DOI or URL of the article must be cited.
License URL: https://creativecommons.org/licenses/by/4.0/

Keywords: Perinatal, Prenatal, Postnatal, Depression, Deep learning, Mental health

Funding: FCT/MECI, Instituto de Telecomunicações UID/50008 FCT/MEC FEDER-PT2020 Partnership Agreement UIDB/00308/2020 This work is funded by FCT/MECI through national funds and when applicable co-funded EU funds under UID/50008: Instituto de Telecomunicações. This work is also funded by FCT/MEC through national funds and co-funded by the FEDER-PT2020 partnership agreement under the project UIDB/00308/2020 (DOI 10.54499/UIDB/00308/2020). The funders had no role in study design, data collection and analysis, decision to publish, or preparation of the manuscript.

==============================
Perinatal depression (PND) refers to a complex mental health condition that can occur during pregnancy (prenatal period) or in the first year after childbirth (postnatal period). Prediction of PND holds considerable importance due to its significant role in safeguarding the mental health and overall well-being of both mothers and their infants. Unfortunately, PND is difficult to diagnose at an early stage and thus may elevate the risk of suicide during pregnancy. In addition, it contributes to the development of postnatal depressive disorders. Despite the gravity of the problem, the resources for developing and training AI models in this area remain limited. To this end, in this work, we have locally curated a novel dataset named PERI DEP using the Patient Health Questionnaire (PHQ-9), Edinburgh Postnatal Depression Scale (EPDS), and socio-demographic questionnaires. The dataset consists of 14,008 records of women who participated in the hospitals of Lahore and Gujranwala regions. We have used SMOTE and GAN oversampling for data augmentation on the training set to solve the class imbalance problem. Furthermore, we propose a novel deep-learning framework combining the recurrent neural networks (RNN) and long short-term memory (LSTM) architectures. The results indicate that our hybrid RNN-LSTM model with SMOTE augmentation achieves a higher accuracy of 95% with an F1 score of 96%. Our study reveals the prevalence rate of PND among women in Pakistan (73.1%) indicating the need to prioritize the prevention and intervention strategies to overcome this public health challenge.

Introduction

Depression has been a significant contributor to non-fatal worsening of health for a period spanning nearly three decades (Davalos, Yadon & Tregellas, 2012). The research findings show that women have a higher likelihood, almost twice as much, of experiencing depression during their lifespan as compared to men (Mbawa et al., 2018). According to the survey, one in seven women experience depression during their lifetime, and most cases occur during the perinatal period (Stein et al., 2014; Kuehner, 2017). PND is more common among pregnant women in some developing countries (Nasreen et al., 2011). Depression rates are higher in low-income countries like 64% in Pakistan (Venkatesh et al., 2016), 18% in Bangladesh (Nasreen et al., 2011), 24.5% in Nigeria (Thompson & Ajayi, 2016), and 24.94% in Ethiopia (Zegeye et al., 2018) and lower in high-income countries such as 7% in Australia (Eastwood et al., 2017), 4.4% in Hong Kong (Lee et al., 2004), and 7.7% in Finland (Pajulo et al., 2001). In India, depression rates range from 91.8% in one study to 36.7% in another study (Ajinkya, Jadhav & Srivastava, 2013). Approximately 45% of Kuwaiti women suffer from postnatal depression (Alenezi et al., 2021).

Women may avoid seeking mental health treatment due to social stigma, lack of knowledge, concerns about the effectiveness of treatment, or unrealistic expectations of motherhood (Lara-Cinisomo, Clark & Wood, 2018). One study suggests screening all women during their perinatal period using standardized, validated tools (Kroenke, Spitzer & Williams, 2001). Perinatal depression (PND) risk factors include lower education, younger age, depression history, domestic violence, financial difficulties, less favorable marital status, negative life events, prenatal depression and anxiety, lack of social support, male gender preference of the child, experiencing a troubled relationship with relatives, and the occurrence of unintended pregnancies (Insan et al., 2022; Schaffir, 2018). Predicting PND in rural and urban areas of a developing country like Pakistan is challenging, but early identification and support can improve outcomes (Waqas et al., 2022).

Researchers have mostly worked with traditional statistical methods and machine learning based methods for detection of PND. Javed et al. (2021) carried out research employing a multi-layer perceptron neural network (MLP-NN) classifier and support vector machine to detect prenatal depression. However, their investigation was confined to analyzing risk factors exclusively in Lahore due to limitations in data availability, which restricts the study’s broader applicability. Liu et al. (2023) identify key limitations in their study: retrospective analysis drawbacks, such as patient selection bias and confounding variables, a focus on cesarean sections requiring further validation for vaginal deliveries, and a reduced sample size due to the use of the propensity score matching (PSM) technique. Huang et al. (2023) reveal that their model’s performance discriminates against women from low-income minority groups. Gopalakrishnan et al. (2022) study’s narrow age range, language requirement for surveys, and tendency to attract healthier participants introduce selection bias, limiting its generalizability and potentially missing women at high risk of PPD. Another study used a small dataset which limits the diversity of algorithms and classifications in ML models (Qasrawi et al., 2022). The researchers gathered incomplete data because participants did not fully complete the online self-reported assessments. The review of these studies reveals that the research community has paid little attention to perinatal depression detection with deep learning (DL) methods.

The following are the major contributions of this study: 1. We have developed a novel dataset named PERI_DEP using Patient Health Questionnaire (PHQ-9), Edinburgh Postnatal Depression Scale (EPDS), and socio-demographic questionnaires, from women participants in the hospitals of Lahore and Gujranwala regions. The PERI_DEP dataset contains 14,008 records.

2. We propose a novel hybrid RNN-LSTM-based deep learning framework for predicting perinatal depression among women in Pakistan. The proposed model achieves an accuracy of 95% surpassing the performance of all the state-of-the-art models previously used for depression detection.

3. To address the class imbalance problem in the dataset, we applied the Synthetic Minority Oversampling Technique (SMOTE) and generative adversarial networks (GAN) oversampling techniques. This approach resolved the class imbalance problem. While using SMOTE improved the performance of the proposed model by a margin of 1%.

4. We have evaluated the integrity of the developed model’s performance by using well-known evaluation metrics including accuracy, precision, recall, F1-score, specificity, area under the curve (AUC), AU-PRC, and Brier score. The proposed model with the SMOTE-based oversampling attains an accuracy of 95% with an AUC of 99%.

The rest of the manuscript is organized as follows: “Related Work” presents an in-depth review of existing work on perinatal depression detection. The methodology of the proposed work is described in detail in “Proposed Methodology”. “Results and Discussion” reports the results and discusses the research findings. Finally, “Conclusion and Future Work” concludes the current study and discusses future work.

Related work

This section presents an overview of the existing work related to the detection of PND. We examine three primary approaches: traditional statistical methods, machine learning, and deep learning models. This comprehensive analysis aims to provide insights into the current state of research on PND prediction and potential future directions in this field.

Traditional statistical methods

A study in rural Bangladesh found 56% prenatal depression rate (Insan et al., 2023), strongly associated with intimate partner violence and husband’s preference for male children. The research involved 235 pregnant women from March to November 2021 and employed statistical analysis, which showed that increased family support reduced depression risk. Likewise, Farooq et al. (2022) conducted a study in Lahore, Pakistan, and found that 82% of pregnant women were depressed—significantly higher than the global average of 10–15%. They used the Hamilton Rating Scale for Depression (HRSD) to diagnose depression, highlighting the severity of the mental health concern among pregnant women in Pakistan. A comparison of depression, stress, and anxiety levels in 180 pregnant women using a cross-sectional design and validated instruments (Doty et al., 2022), found significantly higher abnormal anxiety rates among inpatient (IP) compared to low-risk outpatient (LRO) women.

Machine learning-based models

Jigeer et al. (2022) observed a connection between residential noise exposure during pregnancy and an increased risk of prenatal anxiety and depression in the samples collected from Shanghai, China. The researchers found that depression was more common in 46% of women who worked but quit their jobs due to pregnancy. To analyze the data, the multivariate logistic regression (MLR) model was applied, achieving an accuracy of 71%. This suggested that noise exposure could be a significant factor affecting mental health during pregnancy. Ayyub et al. (2018) found that 39.5% of pregnant women in slum settlements of Lahore suffered from prenatal depression, which was associated with food insecurity. Prenatal depression became a common problem in Pakistan, with a prevalence rate of up to 56% in rural areas. In most cases, intimate partner violence during pregnancy and perceived preference of the husband for male gender of a child, were associated with an increased risk of PND symptoms in Bangladesh (Insan et al., 2023).

Rabinowitz et al. (2023) found that depression was higher in women who were working after childbirth, possibly due to the added stress of balancing a career with caring for a new baby. These multiple roles can lead to role overload, which can hurt a postnatal mother’s psychological well-being. Javed et al. (2021) constructed a multi-layer perceptron neural network (MLP-NN) classifier to identify depression in expected women. They used Relief algorithm to identify features from a dataset of 500 women from Pakistan during prenatal period before training the classifier. The results indicated that MLP and support vector classifiers obtained promising results in identifying prenatal depression with 88.0% and 80.0% and prenatal anxiety with 85.0% and 77.7% accuracy, respectively.

Liu et al. (2023) obtained data from a sample of women who underwent cesarean births where the control population was partitioned into cohorts for training and testing. A total of six distinct machine learning models were used to evaluate their predictive efficacy on the testing population. The SHAP approach was employed to interpret the models. The XGBoost model showed an AUROC of 0.789 in the training and 0.744 in the testing with a threshold of 21.5% for postnatal depression risk probability.

Preis et al. (2022) used machine learning algorithms to analyze data from PROMOTE, a novel psychosocial screening tool for prenatal depression. They found that the model could accurately predict depression with an accuracy of 80%, sensitivity of 75%, specificity of 81%, positive predictive value of 79%, and negative predictive value of 97%. Prabhashwaree & Wagarachchi (2022) used four ML models, Feed-Forward Neural Network (FFANN), Adaptive Neuro-Fuzzy Inference System, Genetic Algorithm (ANFIS-GA), random forest (RF), and support vector machine (SVM) on the dataset containing 686 postnatal data of Sri Lankan women. FFANN model showed the best results with 97.08% accuracy.

Garbazza et al. (2023) carried out a prospective cohort investigation into sleep patterns and mood fluctuations during the perinatal phase. Data was collected from 439 pregnant women during their first trimester, encompassing 47 variables related to socio-demographics, psychology, blood markers, medical/gynecological aspects, and subjective sleep experiences. During the second trimester, data was collected from 353 women, comprising 33 parameters. These two distinct datasets were subsequently subjected to a bivariate analysis to assess their correlations with postnatal depression. SVM was used for predicting early pregnancy PND chances with the sensitivity and specificity of 54.3% and 82.6% respectively. However, polysomnographic (PSG) variables did not enhance the model’s predictive accuracy for PND, instead it relied on subjective sleep issues.

Shin et al. (2020) conducted a study using 28,755 records to assess the effectiveness of nine ML algorithms in predicting postnatal depression. The techniques used in the study were stochastic gradient boosting (GB), recursive partitioning and regression trees, naive Bayes (NB), k-nearest neighbors (kNN), logistic regression (LR), RF, SVM, and neural networks. The RF algorithm attained the highest overall classification accuracy, with a value of 79.1%, and the largest area under the receiver-operating-characteristic curve (AUC) value, which was 0.884. The SVM gained the second-highest performance, with an AUC score of 0.864. Qasrawi et al. (2022) used different ML algorithms such as decision tree (DT), SVM, KNN, RF, NB, and GB. They used a dataset of 3,569 women with the highest accuracy of 83.3% on GB (Goodfellow et al., 2020).

Deep learning-based models

Sharma et al. (2022) conducted a study to figure out stress in pregnant women. They used a deep-learning model based on deep recurrent neural networks (DRNNs) to accurately predict whether working pregnant women were stressed. The proposed DRNN model achieved an accuracy of over 96%. This was a new and improved framework for estimating the stress levels of working pregnant women.

Oğur et al. (2023) used different ML and DL models, including NB, DT, RF, gradient boosting tree (GBT), and deep feedforward neural network (DFFNN). The results showed that the DFFNN model achieved 89.60% higher accuracy with 83.26% precision, 80% recall, and 93.33% F1 score and ML model NB achieved 92.45% accuracy. Table 1 presents an overview of the studies related to perinatal depression.

Table 1 Comparative analysis of existing statistical, machine and deep learning techniques for detection of perinatal depression.

References	Dataset	Sample size	Period	Preprocessing	Data augmentation	Models	Evaluation metrics	Performance	
Oğur et al. (2023)	SUT dataset	250	Perinatal	MVH, N, OR	–	NB	Accuracy	92.45%	
Garbazza et al. (2023)	Multicenter dataset	792	Perinatal	CLN, DF, N	–	SVM	AUC	77.70%	
Huang et al. (2023)	EMR dataset	5,875	Prenatal	DF, N, CV	–	ENet	AUC	61.00%	
Liu et al. (2023)	Cesarean dataset	1,438	Postnatal	CLN, DF, N	–	XGB	AUC	78.90%	
Qasrawi et al. (2022)	Arab dataset	3,569	Perinatal	CLN, DF, N, MVH	–	GB	Accuracy	83.30%	
Sharma et al. (2022)	Pregnant women dataset	–	Prenatal	CLN, DF, N, MVH	–	DRNN	Accuracy	97.00%	
Prabhashwaree & Wagarachchi (2022)	MOH dataset	686	Postnatal	CLN, MVH, CV	–	FFANN	Accuracy	97.08%	
Preis et al. (2022)	EPDS dataset	1,715	Prenatal	MVH, RFE	–	RF	Accuracy	80.00%	
Jigeer et al. (2022)	SFMH dataset	2,018	Prenatal	CLN, DF, N, MVH	–	MLR	Accuracy	71.00%	
Gopalakrishnan et al. (2022)	EPDS dataset	262	Postnatal	MVH, CLN, OR	–	XRT	AUC	81.00%	
Javed et al. (2021)	Pakistani dataset	500	Prenatal	EDA, MVH	–	MLP-NN	AUC	88.00%	
Shin et al. (2020)	PRAMS dataset	28,755	Prenatal	CLN, MVH	SMOTE	SVM	AUC	86.40%	
Note:

None is represented by “-”, MVH, Missing value handling; N, Normalization; OR, Outlier Removal; FD, Filtering Data; CLN, Cleaning; RFE, Recursive Feature Elimination; EDA, Exploratory Data Analysis; CV, Cross Validation.

The existing work utilized datasets having class imbalance problems, which generated biased results affecting the performance of the model (Gopalakrishnan et al., 2022; Andersson et al., 2021). Our study aims to develop a novel dataset named PERI_DEP. In contrast to the previous studies, we utilize SMOTE and generative adversarial networks (GAN) for data augmentation and propose an innovative hybrid RNN-LSTM-based deep learning framework. SMOTE generates new samples by interpolating between existing minority samples, while GAN is a powerful neural network that utilizes a generator and a discriminator in an adversarial setting to learn the data distribution and generate high-quality synthetic data (Goodfellow et al., 2014). We have specifically designed this framework for predicting PND, leveraging the rich dataset we have developed.

Proposed methodology

Figure 1 shows the methodology diagram of the proposed work. The following steps are involved in the methodology.

Figure 1 An overview of the proposed perinatal depression methodology.

Dataset development

Dataset preprocessing

Dealing with class imbalance

Hybrid RNN-LSTM-based DL model training

Model performance evaluation

Dataset development

We gathered data from June 2023 to November 2023 from hospitals in Lahore and Gujranwala regions, employing rigorous questionnaires that included demographic information and mental health assessments (PHQ-9, EPDS). The institutional review board at the Services Institute of Medical Sciences, Lahore Pakistan, approved this study and the approval number was IRB/2023/1155/SIMS. Our PERI_DEP dataset contained 14,008 records from pregnant women and new mothers.

All data was extensively verified and corrected for discrepancies through direct participant interaction to maintain high data integrity. Eligible outpatient department participants were recruited, informed of the study’s purpose, and assured of confidentiality before obtaining their written consent. They then completed questionnaires under the premise of optional participation and data security.

Dataset preprocessing

Current ML and DL algorithms often exhibit a significant decline in performance when confronted with raw data due to their inherent lack of structure and machine-interpretable features. Therefore, preprocessing becomes an essential step in data preparation. In our experiment, data preprocessing includes data cleaning, normalization, one hot encoding, and data labeling. During preprocessing, we identified, 12 samples for which respondents had not provided any response. We removed these samples, leaving 13,996 instances where positive samples are 7,293 and negative samples are 3,903 as shown in Fig. 2A. We used a standard split of 80:20 for training and testing; the training data contains 11,196 instances and the testing split is 2,800, where 1,019 are positive and 1,781 are negative samples as shown in Fig. 2B. As the dataset was imbalanced, SMOTE and GAN oversampling was applied as discussed in the next section.

Figure 2 The class distribution of positive and negative samples before augmentation is shown in (A), and the class distribution of test data is shown in (B).

Dealing with class imbalance

The dataset developed in our study exhibits a notably greater quantity of instances labeled as Depressed compared to those labeled as Non-depressed, leading to an issue of class imbalance. Within the context of this research, a preference for oversampling over undersampling has been made to avoid loss of information and preserve the model’s performance. Data augmentation encompasses the application of SMOTE (Chawla et al., 2002) and GAN (Goodfellow et al., 2014) to create synthetic samples derived from the original dataset. Figure 3 shows the data augmentation using both SMOTE and GAN.

Figure 3 Data augmentation using SMOTE and GAN.

SMOTE generates new synthetic samples for the imbalance class by interpolation between existing samples and their nearest neighbors. While GANs are powerful machine learning models that use two neural networks competing with each other to generate synthetic data that looks real. GAN is better than SMOTE in generating realistic data, handling complex values and diversity, and being capable of generating unlabelled data. The original training data split contains 3,903 non-depressed (negative) cases and 7,293 depressed cases (positive). To balance our dataset, we used both the SMOTE and GAN models, respectively to generate 3,390 samples of non-depressed class. After applying SMOTE and GAN, our training dataset size increased to 14,586. Figure 4 shows that both negative and positive samples have the same number of samples (7,293) in the training data.

Figure 4 Class distribution before/after SMOTE/GAN-based augmentation on the training data.

Hybrid RNN-LSTM-based model training

We proposed a hybrid model combining RNN and LSTM, for higher classification performance to predict PND in women. After performing multiple experiments as discussed in “Results and Discussion”, we found that RNN performed well at capturing short-term dependency in the data. Whereas, LSTM is good at capturing long-term dependencies in complex data by mitigating vanishing gradient problem through its gating mechanism (Sam et al., 2023). In our hybrid RNN-LSTM model, the inputs to the RNN model consist of PERI_DEP preprocessed data. The architecture of RNN includes following layers and components: ‘x’ denotes the input layer, ‘h’ represents the hidden layers, ‘y’ indicates the output layer, and ‘w’ signifies the weights that connect these layers. Each layer functions at a specific time-step ‘t’, reflecting the sequential nature of the data processing in the RNN. The output ‘y’ at any timestamp is computed based on these interactions and the temporal sequence of data, according to Eqs. (1) to (3).

(1) at=b1+Wh(t−1)+Ux(t)

(2) ht=A(a(t))

(3) yt=b2+Vh(t)

where b1 and b2 are bias vectors and ‘W’, ‘U’, and ‘V’ are the weights of layer connections, namely hidden–hidden, input–hidden, and hidden–output, respectively, and ‘a’ is the ReLU Activation function (Gautam et al., 2022). RNNs struggle with issues like vanishing gradients when they need to learn dependencies over long periods. Long short-term memory (LSTM) networks are specifically engineered to overcome these challenges. Each LSTM memory unit comprises a cell state and three distinct gates: the input gate, the forget gate, and the output gate. This structure is crucial for effectively managing information flow, allowing the network to retain or forget data as needed (Uddin et al., 2022).

The input gate It can be as expressed in Eq. (4):

(4) It=β(WLILt+WHLHt−1+b1)

where W is the weight matrix, b is bias vectors, and β is the logistic function. The forget gate Ft, presented in Eq. (5), can be represented as:

(5) Ft=β(WLFLt+WHFHt−1+bF).

The output gate Vt generates the unit’s output and can be described as depicted in Eq. (6):

(6) Vt=β(WLVLt+WHVHt−1+bV).

In this hybrid model, the input layer processes the PERI_DEP dataset, representing numerical records. The initial hidden layer of RNN is designed to capture short-term dependencies, while the subsequent LSTM hidden layer focuses on identifying long-term dependencies. Following these, the network incorporates hidden layers, each with 32 units and employing ReLU activation functions, to decipher complex patterns within the sequential data. The final layer, called the output layer, takes the output from the last hidden layer and maps it to a binary classification task: depressed or non-depressed. To achieve this classification, the output layer employs a sigmoid activation function. Figure 5 illustrates the architecture of the hybrid RNN-LSTM-based deep learning framework used in this study.

Figure 5 Architecture of the hybrid RNN-LSTM model.

Baseline models

To evaluate our hybrid RNN-LSTM model on the PERI_DEP dataset, we compare it with several baseline models. As deep learning baselines, we use CNN-BiLSTM, Thekkekara, Yongchareon & Liesaputra (2024) and LSTM-CNN (Srivatsav & Nanthini, 2024) models. For machine learning baselines, we include well-established algorithms such as random forest (Huang et al., 2023), decision tree (Qasrawi et al., 2022), and gradient boosting tree (Qasrawi et al., 2022). The baselines provide a comprehensive comparison for evaluating the effectiveness of our model.

Results and discussion

Experimental steps

Hardware

The experiments were conducted on a Dell Processor: Intel(R) Core(TM) i5-4210U CPU @ 1.70GHz 2.40 GHz, RAM: 4.00 GB, System type: 64-bit operating system, x64-based processor laptop, catering to the computational needs of the experiments effectively.

Software

Open-source libraries and frameworks are used to implement and train machine learning models and the proposed hybrid model. We used Python’s extensive library ecosystem, including NumPy (https://numpy.org/), Pandas (https://pandas.pydata.org/), and for SMOTE and GAN we used imblearn (https://pypi.org/project/imblearn/) and CTGAN from SDV (https://github.com/sdv-dev/CTGAN) library respectively, to streamline the process. Scikit-learn offered a toolkit for ML tasks, while TensorFlow (https://www.tensorflow.org/resources/libraries-extensions) provided functionalities for deep learning model building and training. All these libraries are freely available online to streamline data pre-processing, model training, and evaluation.

Parameters tuning

We conduct extensive experiments for hyperparameter tuning of each model using grid search to optimize their performance. This involves defining a grid of model-specific hyperparameters, such as the number of hidden layers, learning rate, and regularization parameters, to achieve the best possible results. The grid search method allows us to identify the best-performing hyperparameter configurations by exhaustively testing different combinations and selecting the ones that give the highest performance. We use five dense layers with a batch size of 64 and a learning rate of 0.0001. These hyperparameters are chosen based on the results of these experiments, ensuring the models achieve optimal results. Table 2 shows the values of the hyperparameters.

Table 2 Hyper parameters values.

Parameters	Range	Values	
Dense layers	2–5	5	
Activation function for all layers	–	ReLU	
Activation function for output layer	–	Sigmoid	
Epochs	–	50	
Optimizer	–	Adam	
Batch size	16, 32, 64	64	
Learning rate	0.1–0.0001	0.0001	

Ablation study

We conduct an ablation study that evaluates the performance of various models and training strategies under three scenarios: without augmentation, with SMOTE, and with GAN augmentation as demonstrated in Table 3.

Table 3 Ablation study with SMOTE and GAN and without oversampling.

Models	A	P	R	F1	AUC	AU-PRC	S	B	
Without augmentation	
Only-RNN	0.85	0.84	0.96	0.89	0.95	0.97	0.68	0.10	
Only-LSTM	0.92	0.92	0.96	0.94	0.98	0.99	0.89	0.05	
Hybrid-RNN-LSTM	0.94	0.97	0.94	0.95	0.99	0.99	0.95	0.03	
SMOTE	
Only-RNN	0.90	0.91	0.95	0.92	0.97	0.98	0.81	0.07	
Only-LSTM	0.93	0.95	0.95	0.95	0.98	0.99	0.91	0.04	
Hybrid-RNN-LSTM	0.95	0.95	0.97	0.96	0.99	0.99	0.92	0.02	
GAN	
Only-RNN	0.88	0.87	0.96	0.91	0.96	0.98	0.74	0.10	
Only-LSTM	0.93	0.94	0.95	0.94	0.98	0.99	0.90	0.05	
Hybrid-RNN-LSTM	0.94	0.95	0.96	0.95	0.99	0.99	0.91	0.03	
Note:

Here A, Accuracy; P, Precision; R, Recall; F1, F1-score; S, Specificity and B, Brier score.

Without augmentation: The models are trained directly on the original dataset without any augmentation. Our Hybrid RNN-LSTM model achieves the highest performance across all metrics, showcasing the advantages of combining RNN and LSTM features with an F1-score of 95%. The simple LSTM also performs well, demonstrating its ability to effectively handle sequential data by achieving an F1 score of 94%. In contrast, the performance of RNN models indicates that they struggle with complex patterns. As our dataset is imbalanced, we apply two oversampling techniques for data augmentation.

Data augmentation with SMOTE: First we applied SMOTE to the training dataset. SMOTE generates new instances by interpolating between existing minority samples. After applying SMOTE the results enhance, RNN achieves an accuracy of 90%, while LSTM and our proposed Hybrid RNN-LSTM model archive an accuracy of 93% and 95% respectively. This demonstrates that LSTM maintains consistency while RNN helps mitigate class imbalance, enhancing overall generalization. Our model achieves the best results with SMOTE oversampling.

Data augmentation with GAN: Secondly we applied the GAN oversampling technique on the training dataset, GAN generates realistic synthetic. The hybrid RNN-LSTM consistently performs well, with an accuracy of 94%. This shows our Hybrid RNN-LSTM model performs better on complex time-series and sequential data.

Hybrid RNN-LSTM results

The hybrid RNN-LSTM model achieved an overall accuracy of 95% in predicting depression among women. The model performed well in predicting depression. Figure 6A shows the training and validation accuracy graph, and Fig. 6B depicts the training and validation loss graph. The hybrid RNN-LSTM model shows the best results with the highest accuracy of 95%, 95% precision, 97% recall, 96% F1-score, and 99% AUC.

Figure 6 (A) Shows training and validation accuracy and (B) training and validation loss.

We compare the performance of existing ML and DL models, including RF (Huang et al., 2023), DT (Qasrawi et al., 2022), gradient boosting tree (Qasrawi et al., 2022), CNN-BiLSTM, (Thekkekara, Yongchareon & Liesaputra, 2024) and LSTM-CNN (Srivatsav & Nanthini, 2024) models with our proposed hybrid RNN-LSTM model, utilizing our novel dataset. Although these established ML and DL models showed enhanced accuracy, their performance did not match the superior results achieved by our newly developed model. Table 4 summarizes the performance metrics for all models with the best results highlighted in bold. Figure 7 represents model results before and after data augmentation using SMOTE on the base of the F1-score. The results demonstrate that augmentation positively impacts the performance of all the models. In addition, the proposed model overshadows the performance of all competing models in terms of overall evaluation metrics.

Table 4 Performance metrics of existing and our proposed model using SMOTE oversampling.

Models	A	P	R	F1	AUC	AU-PRC	S	B	
RF (Huang et al., 2023)	0.87	0.92	0.87	0.90	0.87	0.87	0.87	0.13	
DT (Qasrawi et al., 2022)	0.72	0.79	0.75	0.77	0.70	0.75	0.65	0.28	
GBT (Qasrawi et al., 2022)	0.90	0.92	0.87	0.90	0.90	0.87	0.93	0.10	
CNN-BILSTM (Thekkekara, Yongchareon & Liesaputra, 2024)	0.89	0.98	0.85	0.91	0.97	0.98	0.97	0.07	
LSTM-CNN (Srivatsav & Nanthini, 2024)	0.89	0.98	0.84	0.90	0.97	0.98	0.97	0.07	
Hybrid RNN-LSTM (ours)	0.96	0.96	0.98	0.97	0.99	0.99	0.93	0.03	
Note:

Here A, Accuracy; P, Precision; R, Recall; F1, F1-score; S, Specificity and B, Brier score.

Figure 7 Comparison of F1-score on existing models with our proposed model.

According to these results, a large proportion of Pakistani women included in the study sample were facing mental health issues. Several factors were found to be associated with their psychological stress, including young maternal age, low husband family support, low monthly income, having multigravida, and child male gender preference. The findings of this study are concerning, and highlight the need for intervention programs to support women with perinatal psychological distress. This research could help to reduce adverse birth outcomes. Maternity clinics should screen women properly to identify those with symptoms of perinatal psychological distress, allowing for the provision of appropriate counseling and treatment.

Conclusion and future work

The study in hand contributed to the prediction of perinatal depression in three different aspects. First, we have developed a novel dataset (PERI_DEP) by surveying 14,008 participants across various regions of Punjab, Pakistan. The survey showed that 73.1% of women were depressed, while 26.9% of women were non-depressed during the perinatal period. This finding suggests that depression affects a considerable number of women in the studied population, highlighting its prevalence among these individuals. Second, we applied SMOTE and GAN-based oversampling methods for data augmentation. The augmentation had a positive impact on the performance of all the baselines as well as the proposed model. Third, we proposed a hybrid RNN-LSTM model and compared its performance with state-of-the-art models. The proposed model achieved an accuracy of 94% and an impressive F1-score of 96%. One limitation of the study is that the survey data was collected solely from the participants from two cities of Pakistan (Lahore and Gujranwala). This geographically restricted sample limits our ability to extrapolate the findings to the broader perinatal population across Pakistan. To obtain a more comprehensive understanding of PND, future research should consider expanding the scope of data collection to encompass a more diverse geographical representation. This would enhance the generalizability of the results and allow for a more accurate picture of the prevalence and characteristics of PND across a wider population. In addition, future studies could leverage additional data modalities, such as social media data, wearable device data, and biomarker data, to improve the accuracy and predictive power of the proposed model.

Supplemental Information

Supplemental Information 1 Data and code.

Supplemental Information 2 Socio-demographic Questionnaire.

Supplemental Information 3 Patient Health Questionnaire.

Supplemental Information 4 Edinburgh Postnatal Depression Scale Questionnaire.

The authors would like to thank the Services Institute of Medical Sciences, Lahore for granting us the opportunity to collect data for this research. Their cooperation and support were invaluable to the successful completion of this study.

Additional Information and Declarations

Competing Interests

Ivan Miguel Pires and Paulo Jorge Coelho are Academic Editors for PeerJ Computer Science.

Author Contributions

Amna Zafar conceived and designed the experiments, performed the experiments, analyzed the data, performed the computation work, prepared figures and/or tables, authored or reviewed drafts of the article, and approved the final draft.

Muhammad Wasim conceived and designed the experiments, performed the experiments, analyzed the data, performed the computation work, prepared figures and/or tables, authored or reviewed drafts of the article, and approved the final draft.

Beenish Ayesha Akram conceived and designed the experiments, performed the experiments, analyzed the data, performed the computation work, prepared figures and/or tables, authored or reviewed drafts of the article, and approved the final draft.

Maham Riaz conceived and designed the experiments, performed the experiments, analyzed the data, performed the computation work, prepared figures and/or tables, authored or reviewed drafts of the article, and approved the final draft.

Ivan Miguel Pires analyzed the data, prepared figures and/or tables, authored or reviewed drafts of the article, and approved the final draft.

Paulo Jorge Coelho analyzed the data, prepared figures and/or tables, authored or reviewed drafts of the article, and approved the final draft.

Ethics

The following information was supplied relating to ethical approvals (i.e., approving body and any reference numbers):

The institutional review board at the Services Institute of Medical Sciences, Lahore, Pakistan approved this study; the approval number is IRB/2023/1155/SIMS.

Data Availability

The following information was supplied regarding data availability:

The raw data and code are available in the Supplemental File.

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
