# Peer review of "Prediction of perinatal depression among women in Pakistan using Hybrid RNN-LSTM model"

_PeerJ Computer Science, doi:10.7717/peerj-cs.2673_

## Round 0.1 · original submission · Major Revisions

Thank you for submitting your manuscript to PeerJ Computer Science. The review process has been completed, and we have carefully considered the feedback provided by the reviewers.

The reviewers have acknowledged the potential value of your work but have raised several significant concerns. These concerns require substantial revisions to ensure that the manuscript meets the rigorous standards of our journal.

In light of these comments, I am recommending that your manuscript undergoes a major revision. We encourage you to carefully address each of the reviewers’ comments. A detailed response to the reviewers, explaining the changes made or providing justifications for any unaddressed points, should accompany your revised submission.

Once the revisions have been completed, your manuscript will undergo a further round of review to ensure that all major concerns have been satisfactorily addressed.

We appreciate the effort that you have put into this research and look forward to receiving your revised manuscript.

Reviewer 1 ·

Basic reporting

(1) The reviewer recommends the authors to enhance the resolution of the figures.
(2) The reviewer recommends the authors to re-design Table 1, especially the header line of Table 1.

Experimental design

On page 4, Line 186, the authors stated that "applying a GAN oversampling technique to solve the class imbalance problem" and presented the GAN architecture on Figure 3. The reviewer recommends the authors to add more details about "applying GAN oversampling" here, since it is the most important part of this study.

Validity of the findings

The reviewer recommends the authors to also present the performance of the proposed approach without the "GAN oversampling", so the readers can better understand the importance of "GAN oversampling" here.

Additional comments

no comment.

Reviewer 2 ·

Basic reporting

All comments have been added in detail to the last section.

Experimental design

All comments have been added in detail to the last section.

Validity of the findings

All comments have been added in detail to the last section.

Additional comments

Review Report for PeerJ Computer Science
(Prediction of perinatal depression among women in Pakistan using Hybrid RNN-LSTM model)

1. Within the scope of the study, a deep learning-based model was developed to detect perinatal depression using a dataset created specifically for the study.

2. In the introduction section, the importance of depression, perinatal depression and the main contributions of the study were clearly mentioned.

3. In the related works section, studies in the literature that include machine learning-based models, deep learning-based models and traditional statistical methods related to the subject were mentioned. In order to emphasize the importance of the study and its place in the literature more clearly, it is suggested to add columns such as "used dataset, data preprocessing, data augmentation, evaluation metrics" to the literature table.

4. Collecting the dataset used in the study specifically for the study increases the quality and originality of the study in terms of dataset. It was stated that GAN was used to eliminate dataset imbalance. Although there are many different methods that can be used for data augmentation in the literature, it should be stated more clearly why this one was preferred and/or whether different methods have been researched and tried.

5. A deep learning-based hybrid model was proposed within the scope of the study for the detection of depression. Although the architecture of the proposed RNN-LSTM based model has been given, the layers and algorithm of the model should be mentioned in more detail and the originality point should be emphasized more clearly.

6. Since the changes in the parameters such as learning rate, epoch, optimizer given in Table-3 may affect the model results positively/negatively, it should be stated more clearly on what basis these parameters are determined and/or whether different trials are made. Also, since the results are very dependent on the dataset, explain why cross-validation is not preferred for a more accurate analysis.

7. Although metrics such as accuracy and recall have been obtained as evaluation metrics, there are still very serious metrics missing. Obtain the missing metrics by re-examining the metrics obtained in similar studies in the literature.

8. The proposed model results are compared with some other models in Table-5. In this section, it should be explained more clearly on what basis the models selected for comparison were determined. In addition, although the proposed model is a deep learning based hybrid model, the compared models are mostly machine learning models. It is recommended to review state-of-the-art deep learning models in the literature for comparison.

In conclusion, the study is valuable both in terms of the subject addressed and the dataset collected specifically for the study, but it is recommended to pay attention to all the parts mentioned above, such as model details and metrics.

Reviewer 3 ·

Basic reporting

The citations in the text are incorrectly formatted. Each sentence should be able to stand alone without the citations; however, in this instance, removing the citations leaves the sentence without a subject.

Experimental design

The article does not provide information about the sizes of the training, validation, and testing sets, and how those are selected

According to Figure 6 and Table 4, it appears that the authors used the same portion of data for both validation and testing, which is completely incorrect.

Additionally, the description of the usage of GAN suggests that the oversampling technique was also applied to the testing set, which is inappropriate.

Validity of the findings

Because of the incorrect experimental design, the findings are also questionable.

Additional comments

I would like to request authors check other research works on general depression classification (might not perinatal degression), and compare with those methods.

---

## Round 0.2 · Major Revisions

I hope this email finds you well. After a thorough review of your manuscript by the assigned reviewers, I would like to inform you that, while there is potential in your work, several significant concerns have been raised regarding the experimentation and methodology.

The reviewers have pointed out that certain aspects of the experimental setup lack sufficient clarity and justification. In particular, they believe that more detailed explanations and stronger validations are necessary to support your findings. Additionally, methodological improvements have been recommended to ensure the robustness and reliability of the results.

In light of these concerns, we are requesting revisions to the manuscript. We kindly ask that you carefully address each of the reviewers' comments in your revised submission, providing additional detail and supporting evidence where necessary.

Reviewer 1 ·

Basic reporting

no comment

Experimental design

no comment

Validity of the findings

no comment

Additional comments

no comment

Reviewer 2 ·

Basic reporting

All comments have been added in detail to the last section.

Experimental design

All comments have been added in detail to the last section.

Validity of the findings

All comments have been added in detail to the last section.

Additional comments

Review Report for PeerJ Computer Science
(Prediction of perinatal depression among women in Pakistan using Hybrid RNN-LSTM model)

Thank you for the revision. When the changes made to the paper and the responses to the reviewer comments are examined in detail, it is observed that they are at a sufficient level. Best regards.

Reviewer 3 ·

Basic reporting

a) "The results indicate that our hybrid RNN-LSTM model with an augmented dataset achieves a higher accuracy of 98.5% with an F1 score of 98.8%" is unclear and confusing. It looks like the authors might have used augmented data for training, validation, and testing, which needs clarification.

b) The line "The rest of the manuscript is organized as follows: Section ?? presents in-depth review of existing work on perinatal depression detection. The methodology of the proposed work is described in detail in Section ??" contains placeholders ("??") that need correction.

c) figure 7: The Y-axis should be scaled from 0 to 100% instead of 0 to 1 for better readability.

d) "After performing multiple experiments, we found that RNN performed well at capturing short-term dependency in the data. Whereas, LSTM is good at capturing long-term dependencies in complex data by mitigating vanishing gradient problem through its gating mechanism." However, no results were provided for these experiments. The authors should include these results in the main text or add those results to the supplementary section.

e) The comparison of the hybrid RNN-LSTM model should be extended to include more advanced recent models, such as the one mentioned in this paper (https://arxiv.org/abs/2009.12656). The authors should also compare this method with and without GAN data augmentation.

Experimental design

a) The manuscript claims that one of the main contributions is the hybrid RNN-LSTM model. The authors should show comparisons between only RNN, only LSTM, and RNN-LSTM models (with proper hyperparameters tuning) to validate this claim.

b) The manuscript claims that one of the main contributions is the use of GAN for oversampling. The authors need to show the performance of RNN, LSTM, and RNN-LSTM with and without data augmentation, using 2 augmentation techniques (GAN and SMOTE).

c) The hyperparameters provided are the final chosen values, but there is no mention of the considered values for tuning or the method used (e.g., grid search, manual search). This detail should be included for better reproducibility.

d) Cross-validation (e.g., 10-fold) MUST be used for evaluating the model performance due to the small size of the test set. Must provide the standard error with the mean performance.

e) The authors should provide the size of the validation set and its class distribution, as well as the class distribution of the test set.

f) Table 1 contains terms like AUC, AUROC, and ROCAUC. The authors should clarify whether these are different metrics or synonymous.

Validity of the findings

a) The final test performance (98% accuracy) seems suspicious, as the validation accuracy is only around 86%. The authors need to explain this discrepancy.

b) The authors should present performance metrics for their models, including AUROC, AUPRC, F1 Score, Specificity, Recall, Precision, Accuracy, Brier Score, true positive, true negative, false positive, and false negative rates.

Additional comments

a) The DFFNN baseline model is quite basic. The authors should compare their method with more advanced, recent models for a comprehensive evaluation.

b) The manuscript states that GAN-based oversampling was used, but many studies in the literature have already implemented this technique. The authors should discuss what makes their use of GAN unique.

c) The authors should consider revising and clearly presenting the results of other oversampling methods if they tested them, as it is mentioned that GAN was chosen without comparison.

---

## Round 0.3 · accepted · Accept

I hope this message finds you well. After carefully reviewing the revisions you have made in response to the reviewers' comments, I am pleased to inform you that your manuscript has been accepted for publication in PeerJ Computer Science.

Your efforts to address the reviewers’ suggestions have significantly improved the quality and clarity of the manuscript. The changes you implemented have successfully resolved the concerns raised, and the content now meets the high standards of the journal.

Thank you for your commitment to enhancing the paper. I look forward to seeing the final published version.